# A VO_2_ Neuristor Based on Microstrip Line Coupling

**DOI:** 10.3390/mi14020337

**Published:** 2023-01-28

**Authors:** Haidan Lin, Yiran Shen

**Affiliations:** Institute of Modern Circuits and Intelligent Information, Hangzhou Dianzi University, Hangzhou 310018, China

**Keywords:** neurons, VO_2_ Mott memristor, microstrip line

## Abstract

The neuromorphic network based on artificial neurons and synapses can solve computational difficulties, and its energy efficiency is incomparable to the traditional von Neumann architecture. As a new type of circuit component, nonvolatile memristors are very similar to biological synapses in structure and function. Only one memristor can simulate the function of a synapse. Therefore, memristors provide a new way to build hardware-based artificial neural networks. To build such an artificial neural network, in addition to the artificial synapses, artificial neurons are also needed to realize the distribution of information and the adjustment of synaptic weights. As the VO_2_ volatile local active memristor is complementary to nonvolatile memristors, it can be used to simulate the function of neurons. However, determining how to better realize the function of neurons with simple circuits is one of the current key problems to be solved in this field. This paper considers the influence of distribution parameters on circuit performance under the action of high-frequency and high-speed signals. Two Mott VO_2_ memristor units are connected and coupled with microstrip lines to simulate the Hodgkin–Huxley neuron model. It is found that the proposed memristor neuron based on microstrip lines shows the characteristics of neuron action potential: amplification and threshold.

## 1. Introduction

The most important cell in a biological neural network is the neuron. The research of neuroscientists Hodgkin and Huxley pointed out that the processing and sending of nerve pulses by biological neurons are related to the changes in the conductance of sodium ion channels and potassium ion channels in the neuron cell membrane [1,2]. This theory is called the HH model, for which Hodgkin and Huxley won the 1963 Nobel Prize in Physiology or Medicine.

The information processing process of neurons involves the dynamic changes of the membrane potential of neurons. Only when the current neurons stimulate the subsequent neurons to a sufficient extent or frequency will the membrane potential exceed its threshold to stimulate neurons to discharge. If the cell membrane potential does not reach the threshold value, it will automatically decay to the initial state, which is the function of leakage integrated and fire (LIF) of neurons [3]. The relevant “all or nothing” law indicates that the response intensity of nerve cells or muscle fibers does not depend on the intensity of stimulation. The nerve or muscle fiber will be activated if the stimulation is higher than a certain threshold. According to the “all or nothing” law, a single neuron or muscle fiber either responds completely or does not respond at all. If the stimulus is strong enough, an action potential will be generated, and the neuron will transmit information down the axon, away from the cell body, and then to the synapse. The change of cell polarization leads to signal propagation along the length of the axon. The action potential is always a complete response. There is no so-called “strong” or “weak” action potential. On the contrary, this is an all-or-nothing process that can minimize the possibility of information loss during transmission. This process is similar to the action of pressing the trigger of a gun. It is not enough to put a little pressure on the trigger because the gun will not fire. However, the gun will trigger when enough pressure is applied to the trigger. The speed and strength of bullets are not affected by the strength of the trigger-pulling. The gun either fires or does not fire. In this analogy, stimulation represents the force applied to the trigger, while firing represents the action potential.

In recent years, neural computing has been on the rise. The development of special neural computing acceleration chips, such as tensor processing units (TPU), has greatly improved computing efficiency and reduced power consumption. Compared with software-based neural networks, hardware-based neural networks have the advantages of parallel operation and low power consumption [4]. However, traditional electronic devices used to simulate neural synapses and neurons require many components, and the circuit is complex, making it difficult to integrate on a large scale. Determining how to realize the function of neurons with simple circuits is one of the key problems to be solved in this field [5,6,7]. As new circuit elements, one single nonvolatile and one volatile memristor can simulate the functions of synapses and neurons, respectively. Therefore, memristors provide a new way to build hardware-based artificial neural networks [8,9,10].

The innovation and contribution of this paper are to present a new neuristor topology incorporating a microstrip line. The reason to adopt the microstrip line is primarily that inductors are hard to integrate into chips, and when the oscillation frequency is above GHz, distribution effects must be considered. The best choice is the microstrip line. By taking the microstrip line as a building element, our neuristor has a relatively simple topology and is easy to integrate on a chip.

## 2. Hodgkin–Huxley Model

The Hodgkin–Huxley model was described by Alan Hodgkin and Andrew Huxley in 1952. By measuring the response of action potential to different currents in giant squid axons, the two men fitted a model without a computer at that time. This model consists of a set of four ordinary differential equations. Although the description is accurate, the calculation is complex. 

The Hodgkin–Huxley model builds neural cells into circuits, as shown in Figure 1. The lipid bilayer is expressed as capacitance $C_{m}$; the sodium and potassium ion channels are represented by a voltage source $E_{n}$ and conductance $g_{n}$, respectively; the leakage channel is also represented by a voltage source $E_{L}$ and conductance; the resulting ion pump is represented by the current $I_{p}$; the membrane potential is expressed by voltage $V_{m}$. It is worth mentioning that the conductance of sodium and potassium ion channels is related to time and voltage, while the conductance of leakage channels is different from them and is a constant value.

From the circuit formula, we can simply know that the current flowing through the lipid bilayer is
(1)$$I_{c}=C_{m}\frac{dV_{m}}{dt}$$

According to Ohm’s law, the current of sodium and potassium ion channels can be expressed as
(2)$$\begin{array}{l} I_{Na}=g_{Na}\cdot (V_{m}-V_{Na})\\ I_{K}=g_{K}\cdot (V_{m}-V_{K}) \end{array}$$

Of course, because the distribution of ions inside and outside the membrane is different, there are more sodium ions outside the cell and more potassium ions inside the cell, so the directions of the two currents here must be different, and their values must be opposite in sign. The formula of leakage current is the same as both, so it will not be shown.

Thus, it can be obtained that the current flowing through the lipid bilayer is expressed as
(3)$$I=C_{m}\frac{dV_{m}}{dt}+g_{Na}\cdot (V_{m}-V_{Na})+g_{K}\cdot (V_{m}-V_{K})+g_{L}\cdot (V_{m}-V_{L})$$

From the above, we can understand the expression of the current magnitude flowing through the lipid bilayer. Hodgkin and Huxley developed a model through a series of experiments in which the characteristics of excitable cells are expressed by a set of four ordinary differential equations.
(4)$$\begin{array}{l} I=C_{m}\frac{dV_{m}}{dt}+{\bar{g}}_{K}\cdot n^{4}\cdot (V_{m}-V_{K})+{\bar{g}}_{Na}\cdot m^{3}\cdot h(V_{m}-V_{K})+{\bar{g}}_{L}\cdot (V_{m}-V_{L})\\ \frac{dn}{dt}=\alpha_{n}(V_{m})(1-n)-\beta_{n}(V_{m})n\\ \frac{dm}{dt}=\alpha_{m}(V_{m})(1-m)-\beta_{m}(V_{m})m\\ \frac{dh}{dt}=\alpha_{h}(V_{m})(1-h)-\beta_{h}(V_{m})h \end{array}$$
where *I* is the current flow through the unit area. In this neuron model, each ion channel is controlled by multiple subunits. The *n*, *m*, and *h* in the formula correspond to the activation probability of potassium ion channel subunits, the activation probability of sodium ion channel subunits, and the deactivation probability of sodium ion channel subunits, respectively. Because in squid axons, for example, potassium ion channels are controlled by four subunits, the formula is multiplied by the fourth power of *n*, and the other powers are the same. It represents the maximum conductivity of the corresponding ion channel. After multiplying by the corresponding probability, the conductivity at the corresponding time is obtained.

The last three equations $\alpha_{i}$ represent the rate of a subunit from deactivation to activation and represent the rate of a subunit from activation to deactivation. They are related to the membrane potential at this moment. Multiplying the ratio of corresponding activated subunits and inactivated subunits obtains the change.

Through these four equations, we can more accurately express the biological characteristics of membrane potential. In the author’s original paper, the expressions of α and β are given.
(5)$$\begin{array}{l} \alpha_{n}=\frac{0.01(10-V)}{e^{\frac{10-V}{10}}-1}\\ \alpha_{m}=\frac{0.1(25-V)}{e^{\frac{25-V}{10}}-1}\\ \alpha_{h}=0.07e^{-\frac{V}{20}}\\ \beta_{n}=0.125e^{-\frac{V}{80}}\\ \beta_{m}=4e^{-\frac{V}{18}}\\ \beta_{h}=\frac{1}{e^{\frac{30-V}{10}}+1} \end{array}$$
where $V=V_{rest}-V_{m}$.

For the equation of the form such as $\frac{dx}{dt}=\alpha_{x}(1-x)-\beta_{x}x$, we can obtain its solution as follows:(6)$$x(t)=x_{0}-(x_{0}-x_{\infty })(1-e^{\frac{t}{\tau_{x}}})$$
where $x_{\infty }=\frac{\alpha_{x}}{\alpha_{x}+\beta_{x}},\tau_{x}=\frac{1}{\alpha_{x}+\beta_{x}}$.

It is easy to know that when *t* = 0, $x_{0}=\frac{\alpha_{x}}{\alpha_{x}+\beta_{x}}$.

Of course, to make this model run smoothly, we need other parameters. The measured values of these parameters, are also given in the original paper, as shown in Table 1. Respectively, they correspond to the capacitance of the lipid bilayer ($\mu F/cm^{2}$), the resting potential of three ion channels (mV), the maximum conductivity of three ion channels ($mS/cm^{2}$), and the static membrane potential (mV).

Then, through simulation, we can see when the applied pulse current value is less than the threshold and the pulse height is 2 mA, the output voltage waveform is shown in Figure 2. It can be seen from the figure that the output voltage value is very small; almost nothing appears.

When the applied pulse current value is greater than the threshold and the pulse current is 3 mA, the output voltage waveform is shown in Figure 3. It can be seen from the figure that the output voltage suddenly increases.

## 3. Modeling of VO_2_ Local Active Memristor

We adopt the Mott active memristor model and parameters of Wei Yi et al. [11]:(7)$$\begin{array}{l} v=R_{ch}(u)\cdot i\\ \dot{u}={\left(\frac{d\Delta H}{du}\right)}^{-1}\cdot (R_{ch}(u)i^{2}-\Gamma_{th}(u)\Delta T)\\ R_{ch}(u)=\frac{\rho_{ins}L}{\pi r_{ch}^{2}}{\left[1+(\frac{\rho_{ins}}{\rho_{met}}-1)u^{2}\right]}^{-1}\\ \Gamma_{th}(u)=2\pi L\kappa {\left(\ln\frac{1}{u}\right)}^{-1}\\ \left(\frac{d\Delta H}{du}\right)(u)=\pi Lr_{ch}^{2}[{\hat{c}}_{p}\Delta T\frac{1-u^{2}+2u^{2}\ln u}{2u{\left(\ln u\right)}^{2}}+2\Delta {\hat{h}}_{tr}u] \end{array}$$

The first equation is Ohm’s Law, which depends on the state variable, and the second equation is the state equation. The parameters of the equations are listed as Table 2:

We use the Simscape of Matlab to write the script as Figure 4:

## 4. Microstrip Lines

After the 1950s, people began to develop planar transmission lines to meet the needs of electronic technology miniaturization and weight reduction. The planar transmission line is a new type of transmission line following the metal waveguide and coaxial line. It has a planar structure and is small, lightweight, and reliable in performance. It can enable the integration of circuits and systems in RF and microwave bands. There are many types of planar transmission lines. Stripline and microstrip lines are two types of planar transmission lines. The stripline was invented in 1955, and the microstrip line was invented in 1952. In 1965, the first microwave-integrated circuit appeared when solid-state devices were combined with microstrip lines. In RF circuits, planar transmission lines are widely used. Most RF circuits are realized by microstrip lines.

The structure of the microstrip line is shown in Figure 5. It is consist of a conductor strip with a width of *W* and thickness of *t* on one side of the dielectric substrate with a thickness of *h* and a grounding conductor plate on the other side. The overall thickness is very small. Because there is air around the conductor strip of the microstrip line, the microstrip line cannot transmit TEM waves but only quasi-TEM waves. The transmission characteristics of a single microstrip line are approximately treated as TEM transmission lines.

The difference between traditional circuit theory and transmission line theory mainly lies in the relationship between circuit size and working wavelength. In the circuit theory, the size of the network and line is much smaller than the working wavelength, so the amplitude and phase changes of the voltage and current at each point along the line can be ignored. The voltage and current along the line are only related to the time factor but not the spatial location, which is consistent with the basic circuit theory. The voltage and current (or electric field and magnetic field) at each point along the transmission line change with time and space, which is a function of time and space. The voltage and current on the transmission line show volatility.

For a uniform transmission line, because the distribution parameters are uniformly distributed along the line, only the line elements Δz need to be considered. Let the voltage and current at *z* point of the transmission line be v(z,t) and i(z,t)*,* respectively, and the voltage and current at z+Δz are v(z+Δz,t) and i(z+Δz,t), respectively. The line element Δz can be regarded as a lumped parameter circuit. The illustration of the transmission line and its model are shown in Figure 6 and Figure 7 respectively.

The equivalent circuit of the transmission line is as follows:

**Figure 7 micromachines-14-00337-f007:**
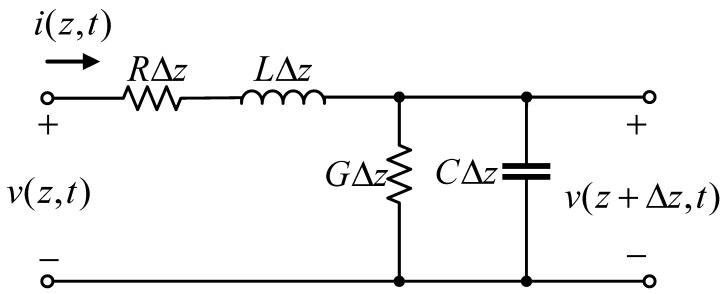
Schematic diagram of transmission line equivalent circuit.

Where *z* is the line length coordinate.

The time domain equation of voltage and current is
(8)$$\begin{array}{l} \frac{\partial v(z,t)}{\partial z}+Ri(z,t)+L\frac{\partial i(z,t)}{\partial t}=0\\ \frac{\partial i(z,t)}{\partial z}+Gv(z,t)+C\frac{\partial v(z,t)}{\partial t}=0 \end{array}$$

The important characteristic parameters related to transmission lines are characteristic impedance and transmission delay. The characteristic impedance of the transmission line is
(9)$$Z_{0}=\sqrt{\frac{R+j\omega L}{G+j\omega C}}$$

The phase velocity is
(10)$$v_{p}=\underset{\Delta t\to 0}{\lim}\frac{\Delta z}{\Delta t}=\frac{c}{\sqrt{\varepsilon_{r}\mu_{r}}}$$
where *c* is the speed of light, $\varepsilon_{r}$ is the relative permittivity, and $\mu_{r}$ is the relative permeability. It can be obtained that the transmission line delay is the characteristic impedance is related to the width of the microstrip line, and the delay is related to the length of the microstrip line, so the microstrip line can be modeled with the characteristic impedance and delay.

## 5. The Artificial Neuristor

It is well known that the two terminal devices using Mott insulators, such as VO_2_, exhibit current-controlled negative differential resistance, which is commonly referred to as threshold switches [12]. This phenomenon is caused by the reversible insulator-metal phase transition. This phenomenon occurs when enough current passes through the device to heat some materials above their phase transition temperature locally, thus forming a conductive channel connecting two electrodes in the device. Because it takes a period to inject enough energy into these devices to heat the materials to the conductive state, they are resistive dynamic systems with a history-dependent excitation. Given this hysteresis and the purely dissipative properties of these devices, the memristor Equation (7) describes their dynamic characteristics.

Mott memristor and Hodgkin–Huxley ion channels have similar dynamic resistance behavior in function, indicating that the former can simulate axonal action potential, especially in the form of artificial neurons. Artificial neuron captures the basic feature of action potential calculation: threshold-driven spike. The lossless peak propagates at a constant speed, with a uniform peak shape and refractory period. From a technical point of view, neurons based on Mott memristors are very interesting because they have a fast conversion speed (<1 ns), low conversion energy (<100 fJ), are at least tens of nanometers in size, are compatible with traditional CMOS materials and processes, and can be manufactured on any substrate. These characteristics are in sharp contrast to the previously proven neurons based on voltage-controlled negative differential resistance devices (e.g., Esaki diode) because that design requires an inductor to work, so it cannot be integrated on a nanoscale.

The memristor neuron circuit introduced here (Figure 8) uses two identical Mott memristors. These two channels are energized (DC bias), with opposite polarity voltages, similar to the sodium channel and potassium channel of the Hodgkin–Huxley model, and are coupled with each other through microstrip lines. The memristor neuron circuit can be regarded as two coupled Pearson Anson oscillations, which are energized below its oscillation threshold (DC stable) and activated by sufficient disturbance (AC unstable) on the input node.

To explain the working mechanism of memristor neurons, we simulate two different input pulses to illustrate the full or no action potential response of the circuit. These inputs are voltage pulses as the stimulation of action potentials generated by upstream neurons. The simulation waveform shows above the threshold (0.3 V 1 ns) and sub-threshold (0.2 V 1 ns). The above-threshold pulse excited the action potential with an amplitude of 0.6 V, while the subthreshold pulse attenuated to 0.05 V, indicating two important bionic characteristics: signal gain and threshold.

Figure 9 is the simulation setup, devices names have been marked in the diagram. The microstrip line is modeled by characteristic impedance and delay time. The transmission delay is set to 5.5 ns, and the characteristic impedance is set to 50 Ώ, respectively. The voltage source in the left leg is set to −2.5 V, while the right leg is set to 0.8 V. There are two voltage sensors that sense input voltage and output voltage, respectively, and send the signals to the Matlab workspace. It should be noted that before sending the signals to the workspace, they have been converted from physical signal format to Simulink signal format.

When the input voltage is 0.2 V, less than the threshold voltage, the input and output waveforms are in Figure 10. The brown line represents the input voltage pulse, while the blue one represents the output voltage (action potential).

It can be seen that the output voltage value is very small and degenerated. The situation is much like that in Figure 2. It shows that when the input voltage is less than the threshold, almost nothing outputs.

When the input voltage is 0.3 V higher than the threshold voltage, the input and output waveforms are shown in Figure 11.

It can be seen that the output waveform suddenly boosts above the original input voltage and, hence, shows some gain. The situation is much like that shown in Figure 3.

From Figure 10 and Figure 11, we can see the designed neuristor shows all-or-nothing properties much like that found in the classical Hodgkin and Huxley model.

## 6. Conclusions

Since the oscillation period of the VO_2_ Mott memristive oscillator is less than 1 ns and, thus, the oscillation frequency is greater than 1 GHz, it has entered the radio frequency range. At this time, the influence of distributed parameters on the circuit performance should be considered. In this paper, we propose a new type of artificial memristive neuristor based on the microstrip line. We have demonstrated the bionic properties of artificial neurons based on nanometer Mott memristors, including the all-or-nothing characteristics of the action potential. In this design, Mott memristors are analogs of ion channels in biological neurons. The designed artificial neuron is compatible with the existing IC technology and materials and can be used to realize transistor-free circuits and new computers based on artificial neural networks. By building and testing the bionic system, we can further understand the signal and information transduction in the artificial neural system.

## Figures and Tables

**Figure 1 micromachines-14-00337-f001:**
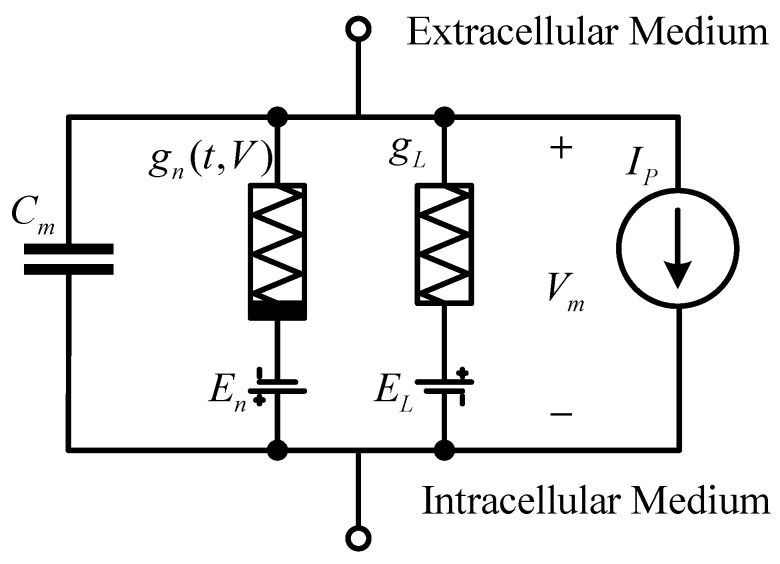
The Hodgkin–Huxley model.

**Figure 2 micromachines-14-00337-f002:**
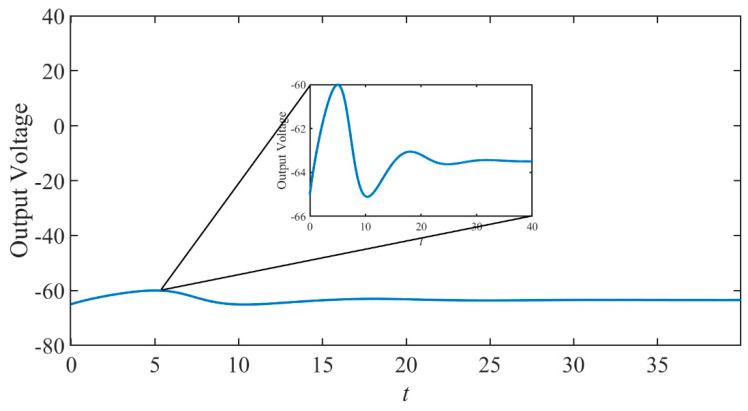
The response is when the input is less than the threshold.

**Figure 3 micromachines-14-00337-f003:**
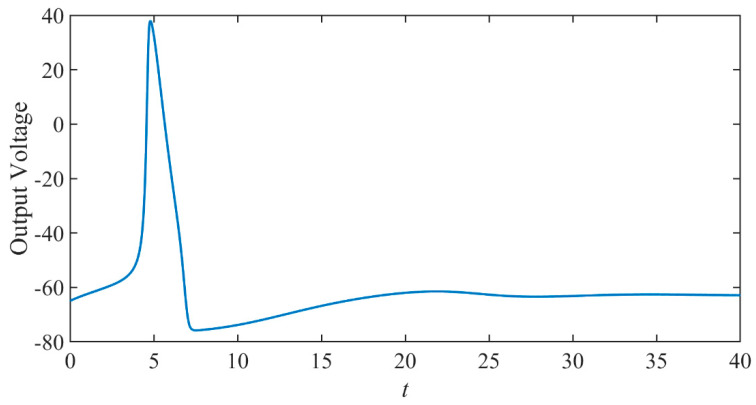
The response is when the input is greater than the threshold.

**Figure 4 micromachines-14-00337-f004:**
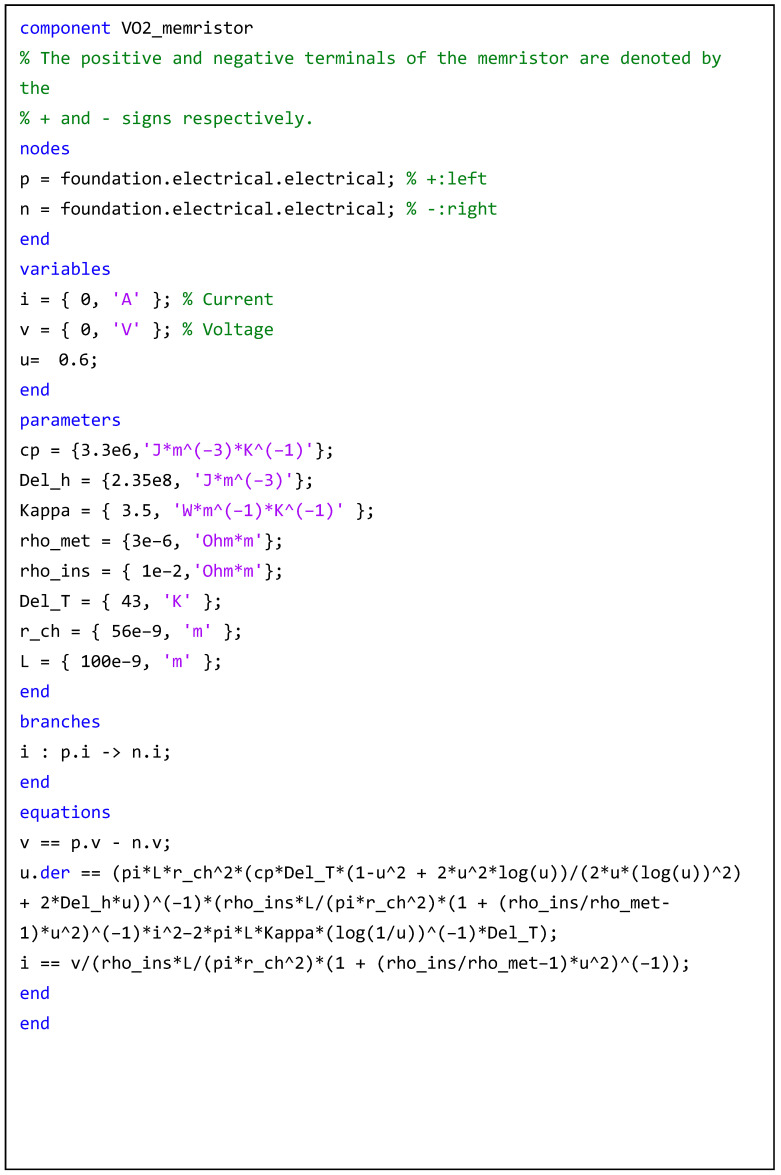
The script of the Simscape of Matlab.

**Figure 5 micromachines-14-00337-f005:**
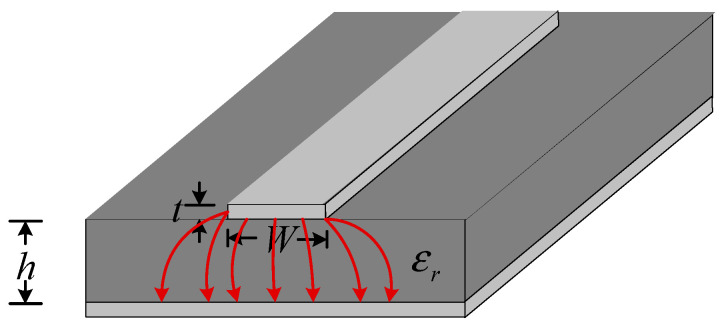
The illustration of the microstrip line.

**Figure 6 micromachines-14-00337-f006:**
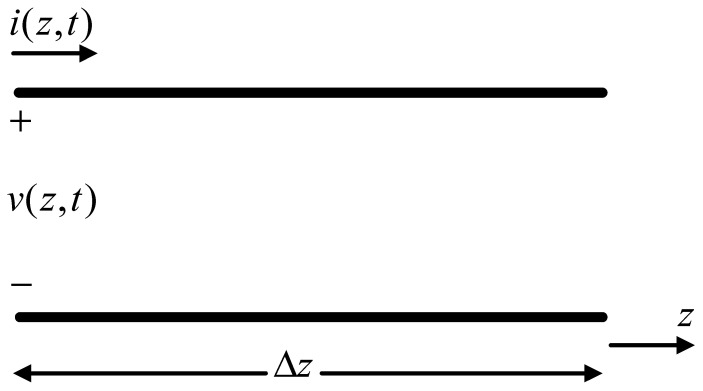
The illustration of the transmission line.

**Figure 8 micromachines-14-00337-f008:**
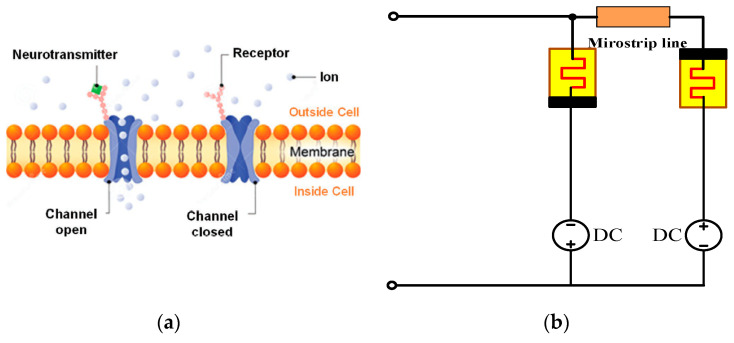
The neuron and the VO2 Neuristor Based on Microstrip Line Coupling. (**a**) The Illustration of neuron [13]; (**b**) The artificial neuristor model.

**Figure 9 micromachines-14-00337-f009:**
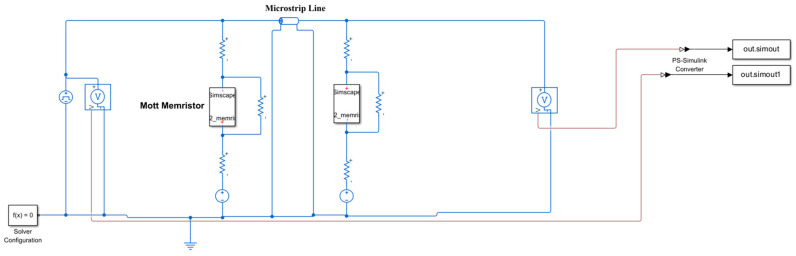
Building an Artificial Neural Circuit Based on Microstrip in Simscape.

**Figure 10 micromachines-14-00337-f010:**
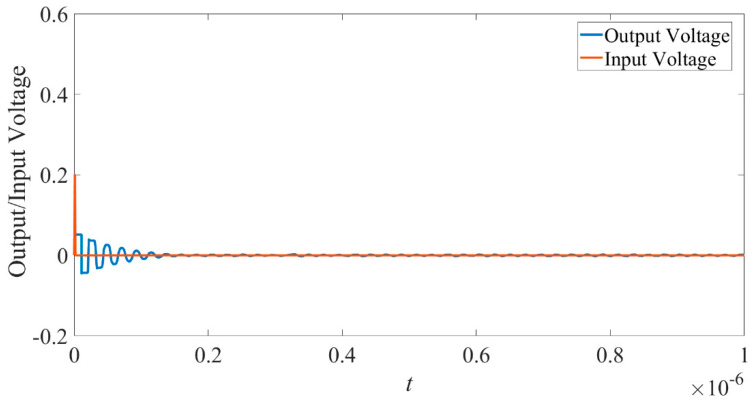
Subthreshold response of the neuristor.

**Figure 11 micromachines-14-00337-f011:**
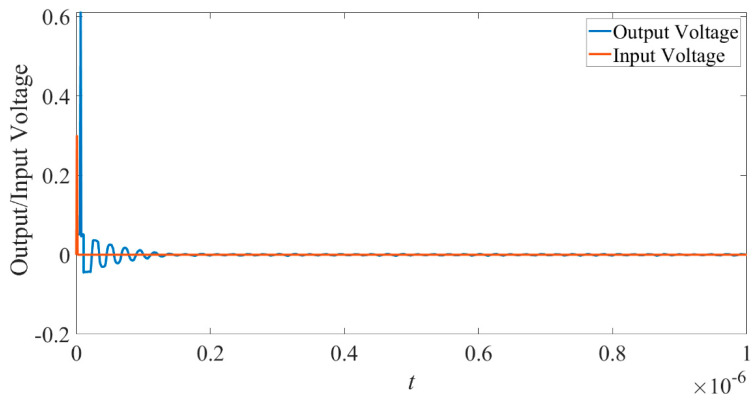
Superthreshold response of the neuristor.

**Table 1 micromachines-14-00337-t001:** The parameters of the model.

$C_{m}$	$E_{Na}$	$E_{K}$	$E_{L}$	${\bar{g}}_{Na}$	${\bar{g}}_{K}$	${\bar{g}}_{L}$	$V_{m}$
1	50	−77	−54.4	120	36	0.3	−65

**Table 2 micromachines-14-00337-t002:** The Parameters of Equation (7).

Memristor Model Property	Symbol	Model Value
Volumetric heat capacity	${\hat{c}}_{p}$	$3.3\times 10^{6}\;J\cdot m^{-3}\cdot K^{-1}$
Volumetric enthalpy of transformation	$\Delta {\hat{h}}_{tr}$	$2.35\times 10^{8}\;J\cdot m^{-3}$
Thermal conductivity	κ	$3.5\;W\cdot m^{-1}\cdot K^{-1}$
Metallic phase electrical resistivity	$\rho_{met}$	$3\times 10^{-6}\;\Omega \cdot m$
Insulating phase electrical resistivity	$\rho_{ins}$	$1\times 10^{-2}\;\Omega \cdot m$
Heating temperature	ΔT	43 K
Conduction channel radius	$r_{ch}$	56 nm
Conduction channel length	L	100 nm
Series electrode resistance	$R_{e}$	300 Ω
Parallel leakage resistance	$R_{shunt}$	15 kΩ

## Data Availability

Not applicable.

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
