# Peer review of "A VO2 Neuristor Based on Microstrip Line Coupling"

_micromachines, 2023, doi:10.3390/mi14020337_

Round 1

Reviewer 1 Report (Previous Reviewer 2)

The paper can be accepted in its present form

Reviewer 2 Report (Previous Reviewer 3)

The authors have responded all my concerns. The paper can be published in its current form.

This manuscript is a resubmission of an earlier submission. The following is a list of the peer review reports and author responses from that submission.

Round 1

Reviewer 1 Report

The potential of Mott VO2 volatile local active memristor to mimic the Neuron has been high lighted by Wei Yi (11) in Nature Communication in 2018. In the present paper two Mott VO2 memristor units are connected and coupled with microstrip lines to simulate the Hodgkin-Huxley neuron model. The implementation of Neuron with two memristor in cross couple have been reported by many authors. For example see  (a).  However it is not clear why in this paper a microstrip line is proposed between two cross-coupled memristors. Besides Si integrated chips are operating at many GHz frequency with copper and low-k interconnects whereas the authors reported simulation frequency is 1 GHz. The author has described a lot on the Hodgkin-Huxley model and the microstrip transmission line while it is hard to find the contribution of the author compared to recent literature. The author should review recent literature on the latest progress on Neuromorphic computation. Some relevant are given below (b) –(c):  

(a)   M. Elhamdaoui, K. Mbarek, S. Ghedira, F. O. Rziga, and K. Besbes, “Synapse design based on memristor,” in 2020 IEEE International Conference on Design & Test of Integrated Micro & Nano-Systems (DTS), pp. 1–5, IEEE, 2020

(b)   D.-A. Nguyen, X.-T. Tran, and F. Iacopi, “A review of algorithms and hardware implementations for spiking neural networks,” Journal of Low Power Electronics and Applications, vol. 11, no. 2, p. 23, 2021.

(c)   M. M. Adnan, S. Sayyaparaju, S. D. Brown, M. S. A. Shawkat, C. D. Schuman, and G. S. Rose, “Design of a robust memristive spiking neuromorphic system with unsupervised learning in hardware,” ACM Journal on Emerging Technologies in Computing Systems (JETC), vol. 17, no. 4, pp. 1–26, 2021.

Author Response

Dear reviewer,

         In my article, I described a neuristor incorporating a microstrip line. Its function is to mimic the famous Hodgkin-Huxley model, detailed circuit implementation is given. The microstrip line is chosen to consider the inductor effects and the inductor is hard to implement in a microchip. Typically the neuristor can be implemented in GaAs process but in GHz range the microstrip line can also be implemented in CMOS process if the interconnecting line is long enough.

       The listed three articles are not relevant to my research, as far as the authors know, only 

       a) Pickett, M. D., Medeiros-Ribeiro, G. & Williams, R. S. A scalable neuristor built with Mott memristors. Nat. Mater. 12, 114–117 (2013).

       b)Yi, W., Tsang, K.K., Lam, S.K. et al. Biological plausibility and stochasticity in scalable VO2 active memristor neurons. Nat Commun 9, 4661 (2018).

are directly relevant to the research, however, their circuits are different from ours.

Reviewer 2 Report

The paper is devoted to the problem of development of neuristor based on modern nanotechnology. A new type of artificial memristive neuristor based on nanometer Mott memristors is proposed in this paper

Authors successfully exploit the property of Mott memristors:  Mott memristor and ion channels in Hodgkin-Huxley model have similar dynamic resistance behavior.

The simulation was performed to confirm the proposal.

 Disadvantages.

The original Hodgkin-Huxley model is low-frequency model (approximately 1 kHz.). The paper discusses circuit versions with the oscillation frequency exceeding 1GHz. For this reason, it is desired to include the subsection with detailed analysis of frequency limitations of the suggested memristive neuristor.

 The paper is of interest to specialists in the field of neuroscience and can be recommended for publication.

Author Response

Dear reviewer,

      Thank you for your time, in your review, you said you are interested in frequency limitations. In fact, the answer has been given in 

Pickett, M. D., Medeiros-Ribeiro, G. & Williams, R. S. A scalable neuristor built with Mott memristors. Nat. Mater. 12, 114–117 (2013).

In the article, the authors said " From a technological standpoint,
neuristors based on Mott memristors are interesting because they
switch rapidly (<1 ns) with low transition energy (<100 fJ), scale
at least to tens of nanometres, are compatible with conventional
front- or back-end complementary metal–oxide–semiconductor
materials and processes, and can be fabricated on arbitrary
substrates."

hence the oscillation frequency is limited by the switching speed of Mott memristor, which is determined by material and the processing technology to produce the memristor.

Reviewer 3 Report

1.        The innovation and contribution of this paper need further explanation.

2.        The authors should give the specific parameters of the simulation results in Figure 2 and Figure 3. The output voltage in the simulation diagrams also should be explained. There seems to be no variable of output voltage in Formula 4, so how is this voltage obtained?

3.        The author gives the concept and formula of Microstrip lines. What is the application of these formulas to this paper?

4.        The analysis of Figures 9 and 10 is inadequate.

5.        The name of each device should be clearly marked in Figure 8, and the whole simulation construction process should be explained in detail.

6.        Why are the signals generated in Figure 8 and 9 a neural signal rather than a damped attenuation signal? How do they relate to the signals in figures 2 and 3?

Round 2

Reviewer 3 Report

The revised version of the paper has not improved significantly.